# DNA Methylation of Pig *FUT3* Promoter Alters mRNA Expression to Regulate *E. coli* F18 Susceptibility

**DOI:** 10.3390/genes12101586

**Published:** 2021-10-09

**Authors:** Zhengchang Wu, Dongfeng Shi, Jian Jin, Hairui Fan, Wenbin Bao, Shenglong Wu

**Affiliations:** 1Key Laboratory for Animal Genetics, Breeding, Reproduction and Molecular Design of Jiangsu Province, College of Animal Science and Technology, Yangzhou University, Yangzhou 225009, China; zcwu@yzu.edu.cn (Z.W.); shijingyzu@163.com (D.S.); fanhairui0328@163.com (H.F.); wbbao@yzu.edu.cn (W.B.); 2College of Veterinary Medicine, Yangzhou University, Yangzhou 225009, China; jianj1127@163.com; 3Joint International Research Laboratory of Agriculture & Agri-Product Safety, Ministry of Education of China, Yangzhou University, Yangzhou 225009, China

**Keywords:** pig, *FUT3*, expression, DNA methylation, *E. coli* F18

## Abstract

Post-weaning diarrhea (PWD) is frequently associated with *E. coli* F18 infections in piglets. However, the underlying molecular mechanism concerning the resistance of *E. coli* F18 in local weaned piglets in China is not clearly understood. In the present study, by a comparative analysis of the transcriptome, a-1,3-fucosyltransferase (*FUT3*) was evaluated as a key candidate correlated with resistance to *E. coli* F18 in Sutai and Meishan piglets. Functional analysis demonstrated that *FUT3* acts as a key positive regulator of *E. coli* F18 susceptibility in newly food accustomed piglets. However, the core promoter of *FUT3* was present at −500–(−206) bp (chr.2: g.73171117–g.73171616), comprising of 9 methylated CpG sites. Among these, the methylation levels of the two CpG sites (mC-3, mC-5) located in HIF1A and Sp1 transcription factor (TF) considerably associated with mRNA expression of *FUT3* (*p* < 0.05). Our findings indicated that the methylation of mC-3 and mC-5 sites has certain inhibitory effect on *FUT3* expression and promotes the resistance of *E. coli* F18 in piglets. The underlined study may explore *FUT3* as a new candidate target in *E. coli* F18 infection in Chinese local weaned piglets.

## 1. Introduction

DNA methylation widely existing in mammals can regulate gene expression and maintain genetic stability with transcription and cell division. Under the catalysis of DNA methyltransferase (DNMTs), S-adenosyl-L-methionine (SAM) as methyl group is added to DNA segments and is converted to 5-methylcytosine (5-mC), N^6^-methyladenine (m^6^A), and 7-Methylguanine (7-mG). Among these, 5-mC is the most important methylation modification in mammals. DNA methylation is the main epigenetic mechanism in promoter regions of the eukaryotic genomes that regulate the expression of a gene by enhancing or attenuating the interaction of TF with sequences of DNA in the promoter region [1]. It has been reported that DNA methylation plays a crucial role in maintaining cell function, regulating individual development and disease [2,3,4]. DNA methylation is considered to be a significant research hotspot in the current field of pig genetics and breeding, relevant studies mainly focus on tissue-specific expression [5,6], cell apoptosis [7], variety differences [8], growth and development [9,10,11], immune response [12,13]. However, DNA methylation has been rarely reported in expression regulation of pig resistance to pathogenic microorganism infection.

Piglets’ bacterial diarrhea is one of the most common intestinal inflammatory diseases that leads to severe financial damages to pig farming on large scale. Enterotoxigenic *Escherichia coli* F18 (*E. coli* F18) is the main pathogenic microbe that results in post-weaning diarrhea [14]. However, it is evident from the study that receptors for *E. coli* F18 in the small intestine of a piglet at the brush border of the epithelial cell, are potentially associated with the pathogenicity of *E. coli* F18 in terms of their expression levels. In addition, the mechanism being involved in *E. coli* F18 resistance in Chinese local weaned piglets still remains undiscovered. In this study, based on previously obtained Sutai (a new crossbred between Meishan and the Duroc strains) and Meishan piglets, i.e., resistant and sensitive to *E. coli* F18 [15,16], we conducted a comparative mRNA sequencing of duodenum tissues and identified a functioning gene, designated a-1,3-fucosyltransferase (*FUT3*), that may have critically contributed in anti-*E. coli* F18 infection. FUT3, belonging to the fucosyltransferase family, regulates the formation of ABH and Lewis antigens and resistance to pathogen infection [17,18]. To further investigate the mechanism of *FUT3* expression in regulating *E. coli* F18 resistance, this study systematically verified the linkage between the expression of *FUT3* and infection of *E. coli* F18 at the tissue and cellular levels. Then, the *FUT3* core promoter was determined by bioinformatics analysis and dual-luciferase assay, meanwhile, bisulfite amplicon sequencing (BSAS) was used to detect the level of methylation in *FUT3* core promoter in intestinal tissues of piglets, i.e., sensitive and resistant to *E. coli* F18. Then we determined the impact of the methylation sites on the expression of *FUT3* mRNA and evaluated considerable transcription factor in the core promoter region. Hence, this study conformed a candidate target for combating vulnerability to *E. coli* F18 in weaned piglets and further revealed the effect of DNA methylation on gene expression. Our findings aim to provide insights into the molecular mechanism of piglets’ anti-*E. coli* F18 infection and providing the basis for solving the key scientific problem of *E. coli*-resistant disease breeding in Chinese local pigs.

## 2. Results

### 2.1. FUT3 Was Evaluated as a New Target to Combat E. coli F18 Infection on the Basis of Comparative Transcriptome Analysis

To determine the molecular mechanism of regulating *E. coli* F18 resistance in local pig breeds in Chinese indigenous pigs, we conducted a comparative RNA-seq analysis of duodenal tissues from Sutai and Meishan piglets exhibiting sensitivity or resistance to *E. coli* F18. In Sutai piglets, 238 differentially expressed genes (DEGs) were screened out between piglets that were sensitive or resistant to *E. coli* F18 (*p*-value <0.05 and |fold change)| <2 (Figure 1a). Likewise, 310 DEGs were detected in Meishan piglets (Figure 1b). Interestingly, we identified 46 common DEGs between two pig breeds (Figure 1c,d). Among these DEGs, α-1,3-fucosyltransferase (*FUT3*) is involved in glycosphingolipid biosynthesis (KO pathway: ko00601) correlated with the generation of *E. coli* F18 receptor [19,20,21], which is probably considered as a novel target to combat *E. coli* F18 infection in piglets.

### 2.2. Knockdown of Pig FUT3 Enhances E. coli F18 Resistance

To analyze the correlation between the expression of *FUT3* and *E. coli* F18 susceptibility in pigs, we analyzed the differential mRNA and protein expression of *FUT3* between piglets that were sensitive and resistant to *E. coli* F18. qPCR evaluation revealed that the *FUT3* gene’s expression level was considerably elevated in the duodenum and jejunum of piglets that were sensitive to *E. coli* F18 relative to those that showed resistance (*p* < 0.01), as depicted in Figure 2a. The results were further confirmed by the immunoblotting which revealed that the protein expression of *FUT3* was significantly upregulated in a sensitive group, as depicted in Figure 2b.

To further understand how the expression of *FUT3* regulated *E. coli* F18 invasion, the lentivirus-activated RNAi was used to facilitate the attenuation of *FUT3* in the intestinal porcine epithelial cells (IPEC-J2). The expression of a GFP was determined in more than 90% of cells treated with shRNA vector. As indicated in Figure 3a, The knockdown potential of *FUT3* in sh*FUT3*-n (*n* = 1, 2, 3, 4) treated IPEC-J2 cells were 39.9%, 71.6%, 23.1%, and 86.9%, accordingly, relative to non-treated cells (control) and sh*FUT3*-4 treated IPEC-J2 cells were considered for the successive evaluation. The level of Protein expression showed consistency with transcription levels (Figure 3b). Therefore, we successfully established the IPEC-J2 cell line with the silencing of *FUT3*. Besides, we further investigated the effects of *FUT3* expression on the interaction of *E. coli* F18 to IPEC-J2 cells. After *FUT3* knockdown, bacteria enumeration (Figure 3c) and relative quantification (Figure 3d) revealed that the adhesion levels of *E. coli* F18ab/ac-expressing fimbriae to IPEC-J2 cells in the sh*FUT3* group were considerably reduced than that in control groups (*p* < 0.01). Immunofluorescence assay (Figure 3e), scanning electron microscopy (Figure 3f), and gram staining (Figure 3g) also showed that the distribution of *E. coli* F18 in the sh*FUT3* group was markedly higher than that in the control group. These results suggested that *FUT3* contributed to regulating *E. coli* F18 infection and lowered expression level enhances *E. coli* F18 resistance.

### 2.3. Impact of Pig FUT3 Promoter Methylation on Gene Expression

To further investigate the molecular mechanisms of *FUT3* expression, we focused on the DNA methylation modification analysis of pig *FUT3* promoter. Firstly, we carried out the determination of the *FUT3* core promoter region. According to the IGV analysis of transcriptome sequencing, we achieved a 2000 bp sequence upstream of the *FUT3* gene. The prediction of three promoter regions of pig *FUT3* was carried out which were present at –400–(–350)bp, –1297 –(–1247)bp, –1848–(–1798)bp upstream of the transcription start site. Then, the 2000-bp upstream sequence was divided into four fragments, namely, –200–0 bp (control), –500–0 bp (P1), –1500–0 bp (P2), and –2000–0 bp (P3) (Figure 4a). Agarose gel electrophoresis was employed for evaluating the PCR products, followed by sequencing. PCR amplification products were evaluated via agarose gel electrophoresis (1%), as depicted in Figure 4b. As shown in Figure 4c, luciferase assay revealed that the luciferase intensity of the pRL-P1 was considerably elevated (unpaired *t*-test, *p* < 0.01) than that of the other transfected groups. We identified that the core promoter region of *FUT3* was found at –500–(–200) bp (chr. 2: g.73171117–g.73171616). Based on sequence, we designed the methylation amplification region (Figure 4d) using the MethPrimer software.

Furthermore, we conducted the methylation analysis of the *FUT3* promoter in intestinal tissues of individuals that were resistant and sensitive to *E. coli* F18 via bisulfite amplicon sequencing (BSAS). As shown in Figure 5a, we detected nine CpG sites methylated in pig *FUT3* core promoter. On the whole, the average methylation levels in the duodenum (78.25%) and jejunum (86.63%) tissues from *E. coli* F18-resistant individuals were considerably elevated than that in the duodenum (72.20%) and jejunum (75.15%) tissues from *E. coli* F18-sensitive individuals (*p* < 0.05). From single CpG sites, the methylation levels of mC-5 and mC-6 sites in intestinal tissues from individuals that were resistant to *E. coli* F18 have considerably elevated than that from *E. coli* F18-sensitive individuals (*p* < 0.05 or *p* < 0.01, Figure 5b,c). Interestingly, Pearson correlation analysis revealed that the methylation level of the CpG sites remained in a negative correlation with the mRNA expression of *FUT3* (Figure 5d, R = −0.643, *p* < 0.05), with considerable correlation coefficients being attained only for the mC-3 (R = −0.53, *p* < 0.05), mC-5 (R = −0.818, *p* < 0.01) and mC-6 sites (R = −0.83, *p* < 0.01), indicating the methylation of CpG-(3, 5, 6) sites probably inhibit *FUT3* mRNA expression.

### 2.4. Evaluation of Key Transcription Factors in Pig FUT3 Core Promoter

To further identify the important transcription factors in regulating *FUT3* expression, we presented the potential transcription factor binding sites (TFBS) found within CpG sites in the core promoter region of pig *FUT3* genes, such as SP1, HIF1A, AP2, USF, C/EBPα, and CREB (Figure 4d). In the above study, we found a significant correlation between the methylation level of CpG-(3, 5, 6) sites and *FUT3* mRNA expression (Figure 5d). Furthermore, mC-3, mC-5, and mC-6 were located in the HIF1A (hypoxia-inducible factor 1-α), SP1 (specificity protein 1), and USF (upstream stimulatory factor) transcription factor binding site, which reveals their significant contribution in facilitating expression level of *FUT3* by influencing the interaction of TFs with a sequence of the promoter. Herein, we performed dual luciferase activity assay to investigate the effects of HIF1A, SP1, and USF on transcriptional activity in the *FUT3* promoter. As shown in Figure 6, HIF1A and SP1 could promote *FUT3* transcriptional activity, while USF led to the inhibition of transcriptional activity.

## 3. Discussion

To date, no study has assessed the *E. coli* F18 resistance in Chinese local weaned piglets and their molecular mechanism is still remaining elusive. In previous studies, the TLR genes, i.e., *TLR5* and *CD14* play important roles in the pig intestine inflammatory response against invading *E. coli* F18 pathogens [22,23]. In addition to intestinal immunity, resistance to *E. coli* F18 is most commonly correlated with the expression of the level of its receptor in the intestinal epithelial cells in piglets [24]. According to recent studies, F18-fimbriated *E. coli* interacts with the small intestine of piglets by latching onto glycosphingolipids with blood group ABH determinants on the type-1 core [19,20]. ABH and Lewis antigens (which are the types of Histo-blood group antigens, HBGAs) are formed from precursor antigens, followed by modification by various glycosyltransferases, including a-1,2-fucosyltransferase (FUT2), FUT3 [25]. Herein, we have been carried out a comparative transcriptome analysis of tissue samples taken from the duodenum, followed by identification of a-1,3-fucosyltransferase (*FUT3*) as a candidate target to battle the susceptibility to *E. coli* F18 in Chinese local piglets. *FUT3* belongs to the family of fucosyltransferase genes and was found to be associated with the mediators of gastrointestinal disease, rotavirus, and bacterial infection [17,26]. To investigate the effect of *FUT3* expression on *E. coli* F18 resistance in piglets, qRT-PCR and immunoblotting were carried out to validate the differential expression of *FUT3* in intestinal tissues of piglets that were resistant and sensitive to *E. coli* F18. The obtained results indicated a considerably downregulated expression in piglets that were sensitive to *E. coli* F18. Moreover, we performed an in-vitro functional analysis of the *E. coli* F18 adhesion via RNA interference, followed by demonstrating that *FUT3* knockdown markedly enhanced the *E. coli* F18 adhesion level to intestinal porcine epithelial cells (IPEC-J2). Therefore, it can be speculated that *FUT3* expression was involved in the formation of the *E. coli* F18 receptor and functioned as a positive regulator of enhancing the susceptibility to *E. coli* F18 in piglets.

To further extensively evaluate the mechanisms that regulate the *FUT3* expression alterations in *E. coli* F18 infected pigs, the methylation of the *FUT3* core promoter region in the jejunum and duodenum tissues of piglets (sensitive and resistant to *E. coli* F18) were evaluated through methylation sequencing. Promoters are often located at 5′flanking regions, DNA methylation in the regions is considered to be the most obvious epigenetic gene expression regulation. In common, the variations in DNA methylation take place in the CpG [6,11,22], and non-CpG islands [5]. Herein, we examined the 2000 bp upstream of the pig *FUT3* gene and observed that there were no CpG islands in this region. In this view, we evaluated the luciferase assay of the successively 5’-shortened sections of *FUT3* promoter for the determination of the sequence in the core promoter region (–500–(–200)bp). Our studies showed the higher methylation levels in intestinal tissues of piglets (resistant to *E. coli* F18) and were found to be exhibited a significant negative correlation (R = −0.643, *p* < 0.05) between the average methylation level and the expression of mRNA, indicating that *FUT3* methylation may inhibit its expression and enhance the immunity of piglets against *E. coli* F18 infection. Furthermore, multiple studies have revealed that the occurrence of many CpG sites in the promoter region of a particular gene, but some specific CpG sites can affect gene expression [27]. Analyses of single methylation sites identified a significantly negative association in mRNA expression levels among the mC-3, mC-5, and mC-6 sites of *FUT3* core promoter (*p* < 0.05). In view of these observations, the underlined sites might significantly regulate the transcription of the gene. DNA methylation modulates transcriptional factor (TF) activity by regulating TF binding sites (TFBS) located in the regions with specific CpG sites, leading to the activation or inhibition of gene transcription [6]. Based on transcription factor prediction, we investigated several important transcription factors in *FUT3* promoter CpG sites, which belongs to the methylation-dependent transcription factors and include hypoxia-inducible factor (HIF)-1α (HIF1A) [28], specificity protein 1 (SP1) [29,30], cAMP response element-binding proteins (CREB) [31], upstream transcription factor (USF) [32], CCAAT/enhancer-binding protein β (C/EBPβ) [6]. Interestingly, these mC-3, mC-5, and mC-6 sites with inhibiting *FUT3* expression were located in HIF1A, SP1, and USF, respectively. The dual-Luciferase assay showed, HIF1A and SP1 could promote *FUT3* transcriptional activity, while USF caused the inhibition of transcriptional activity. Taken together, we revealed that the elevated level of methylation at mC-3 and mC-5 regions overcome the SP1 and HIF1A binding potencies to the *FUT3* promoter to suppress the gene expression. In future studies, there is a need for validation of HIF1A and SP1 binding to the *FUT3* promoter via electrophoretic mobility shift assays (EMSA) and chromatin-immunoprecipitation (ChIP), meanwhile, we will explore in-depth the function of a key transcription factor by RNA overexpression and CRISPR/CAS9-knockout.

## 4. Conclusions

In conclusion, the present study identified *FUT3* as a key target for combating susceptibility to *E. coli* F18 in Chinese local pigs based on comparative transcriptome analysis. Functional verification indicated that the low expression of *FUT3* was linked with resistance to *E. coli* F18 in piglets. Besides, we determined the *FUT3* core promoter region (–500–(–200 bp) by bioinformatics analysis and luciferase assay. Methylation analysis showed, two key CpG sites (mC-3, mC-5) probably decreased the binding activity of HIF1A and SP1 to the promoter of *FUT3* for the inhibition of gene expression. Our obtained results may explore the regulatory mechanism of the pig *FUT3* gene that affects resistance to the infection caused by *E. coli* F18 and involved in its application in molecular breeding against bacterial diarrhea in local piglets of China.

## 5. Materials and Methods

### 5.1. Ethics Statement

The IACUC of Yangzhou University provided approval for all animals conducted studies, under the permit number: SYXK (Su) IACUC 2012-0029. All piglets related experiments were conducted on the directions of administration and regulatory affairs regarding the approval of Experimental Animals by the State Council of the People’s Republic of China.

### 5.2. Experimental Sample

Experimental pigs (Meishan, Sutai) were acquired from Kunshan Conservation Ltd. and Sutai Pig Breeding Center. In the previous study, the model animals, i.e., weaning piglets were used to check their vulnerability to *E. coli* F18 by challenging with the *E. coli* through feeding strains, i.e., F18ab and F18ac [16]. Based on functional adhesion and receptor binding assays for the type V secretion system established by our research team [33], three pairs of piglets (resistant and sensitive to *E. coli* F18) with similar birth weight, coat color, the shape of the body and weaning weight were chosen for the analysis of RNA-seq. All experimental piglets were euthanized via intravenous injection of pentobarbital sodium and duodenum, liver, heart, spleen, kidney, lung, stomach, kidney, muscle, lymph node, thymus, and jejunum tissues were collected, followed by storing in liquid nitrogen in situ for further use.

### 5.3. Transcriptome Sequencing (RNA-seq)

Total RNA of duodenum tissues from 6 Meishan piglets (Resistant group: MR-1, MR-2, MR-3; Sensitive group: MS-1, MS-2, MS-3) and 6 Sutai piglets (Resistant group: SR-1, SR-2, SR-3; Sensitive group: SS-1, SS-2, SS-3) was isolated by TRIzol™ reagent (Invitrogen, ThermoFisher Scientific, Carlsbad, CA, USA) on the basis of suggestion provided by the manufacturer. The removal of rRNAs from total RNA was carried out via NEBNext rRNA Depletion Kit (USA), as suggested by the manufacturer. Libraries of RNA were formed via the NEBNext^®^ Ultra™ II Directional RNA Library Prep Kit (USA), as suggested by the manufacturer. Libraries quality and quantification were carried out via the BioAnalyzer 2100 system (USA). RNA Sequencing was conducted at the Novogene Bioinformatics Institute (Beijing, China) on an Illumina Hiseq 2500 platform and 100 bp paired-end reads were constructed. The obtained raw RNA-seq data (high-throughput) were submitted to NCBI (BioProject ID: PRJNA476718, PRJNA476720, PRJNA476721, PRJNA476722).

Paired-end reads were collected from Illumina HiSeq 2500 sequencer, and their quality was controlled via Q30. Post 3′ adaptor-trimming and reads (low quality) removal by cutadapt (version 1.9.3), the clean reads with higher quality were aligned to the reference genome (SusScr11) with the software, i.e., hisat2 (version 2.0.4). Then, the cuffdiff software was employed to obtain the gene level FPKM as the expression profiles of mRNA, and a threshold for considerably variant expression was set as a *p* < 0.05 and |log2(fold change)| > 1.

### 5.4. Knockdown Analysis

Four RNAi sequences targeting *FUT3* mRNA (R1, R2, R3, R4) and one off-target control (NC) sequence were designed (Table 1) and individually cloned into LV3-H1/GFP&Puro vector (GenePharma), followed by co-transfection with packaging plasmids into 293T cells (GenePharma). The collection of the virus was carried out that was used for infection of the target cells (IPEC-J2). Positive cells were chosen by adding puromycin (10 μg/mL). After 2 days of transfection, total RNA of various treatment groups was isolated via Trizol. PCR was conducted to determine the target gene expression in cells. Based on *FUT3* expression, the lentivirus with the elevated interference potential was chosen.

### 5.5. Adhesion Level Detection of E. coli F18 to IPEC-J2 In Vitro

Intestinal porcine epithelial cells (IPEC-J2, provided by the University of Pennsylvania, USA) were grown in medium, i.e., DMEM/F12 in a 1:1 ratio provided with FBS (10%). For the IPEC-J2 control (Control), FUT3 knockdown cells (shFUT3), in vitro evaluation of F18ab fimbriae standard strain 107/86 (O139:K12:H1) and F18ac (8199) adhesion was performed as explained earlier [33]. To establish an effective and precise method for the detection of *E. coli* adhesion, a relative quantification method [34], bacteria enumeration, gram staining, scanning electron microscopy (SEM), immunofluorescence assay (IFA) [22] were used for measuring *E. coli* F18 interacting ability with epithelial cells of the small intestine in pigs in vitro.

### 5.6. RT-qPCR Analysis

Total RNA from IPEC-J2 cells and tissues was isolated with the help of Trizol Reagent. The purity, as well as total RNA concentration, were evaluated via agarose gel electrophoresis in formaldehyde (1%) and NanoDrop 1000 (USA), then RNA was kept at −70 °C. Synthesis of cDNA was carried out via 1 µL of 5 × qRT SuperMix II along with total RNA of 500 ng, and 10 µL of RNase free ddH_2_O. RT-qPCR instrument ABI7500 was employed for the evaluation of qPCR and the parameters were set as 95 °C for 30 and 5 s, accordingly, followed by 60 °C for 34 s, and then 40 cycles were carried out. The primers of amplified fragments of *FUT3* and *β-actin* were indicated in Table 2.

### 5.7. Determination of Pig FUT3 Core Promoter Region

The 2000-bp upstream sequence of the transcription start site of pig *FUT3* gene was obtained by NCBI and UCSC database. The prediction of CpG islands of the pig *FUT3* upstream region was performed via the MethPrimer software. Besides, the prediction of the *FUT3* core promoter region was carried out via BDGP software. Based on the above bioinformatics analysis, the 2000-bp upstream sequence was designed for the amplification of various promoter fragments truncated at the 3′ end of the sequence. Then, the resulted PCR products were evaluated via agarose gel electrophoresis, then sequencing was carried out. Recombinant vectors were labeled as *FUT3*-200bp (Control), *FUT3*-500bp (P1), *FUT3*-1500bp (P2), *FUT3*-2000bp (P3), and then co-transfected with pRL-TK vector into cells for dual luciferase assay.

### 5.8. Detection of Methylation in Pig FUT3 Core Promoter Region

Genomic DNA from duodenum and jejunum tissues was transformed via EpiTect Fast DNA Bisulfite kit (QIAGEN, Hilden Germany), and the underlined PCR primers (F:GTTTGAAATTTAAGTTTTATGAATT,R:ATAAAAAATACAACCTCTCCCTACC) were established on the basis of the sequence of the sulfite transformation via the MethPrimer software. Moreover, 25-μL PCR amplification reactions of bisulfite-treated DNA (BST-DNA) consist of 2 μL DNA template, 12.5 μL ZYMO Taq Premix, 1 μL forward primer (10 μM/L), 1 μL reverse primer (10 μM/L), 8.5 μL distilled DI H_2_O. The subsequent reaction conditions were as follows: 95 °C for 10 min, then 35 cycles of 95 °C for 30 s, 52 °C for 30 s, 72 °C for 35 s, and 72 °C for 10 min. The TIANquick Midi purification kit (Tiangen Biotech, Beijing) was utilized for the purification of PCR products (209 bp), followed by cloning into the pMD19-T vector (Takara, Dalian, China). A total of 30 positive clones were randomly chosen for sequencing of all samples (Invitrogen Biotechnology, Shanghai, China) and sequence data for methylation level were evaluated via the quantification tool for methylation analysis (QUMA) software [35].

### 5.9. Identification of Key Transcription Factor in Pig FUT3 Core Promoter Region

Alibaba (http://www.gene-regulation.com/pub/programs/alibaba2/index.html?, accessed on 18 February2021) and JASPAR (http://jaspar.genereg.net/cgi-bin/jaspar_db.pl, accessed on 18 February 2021) were employed to evaluate transcription factor binding sites (TFBS), basing on the sequence of the *FUT3* promoter region. Based on the CpG sites of methylation level significantly altering mRNA expression, we further screened out the important transcription factors and conducted validation experiments by dual-luciferase activity assay.

### 5.10. Statistical Analyses

In order to conduct statistical analyses, the SPSS 18.0 software (Chicago, IL, USA) was used. Relative quantitative results were examined via the 2^−ΔΔCt^ method [36]. Results were represented as mean ± SEM, followed by comparison via Student’s *t*-test. *p* < 0.05 or *p* < 0.01 indicated statistical significance. Pearson’s correlation was performed to evaluate the association between *FUT3* promoter methylation and mRNA expression.

## Figures and Tables

**Figure 1 genes-12-01586-f001:**
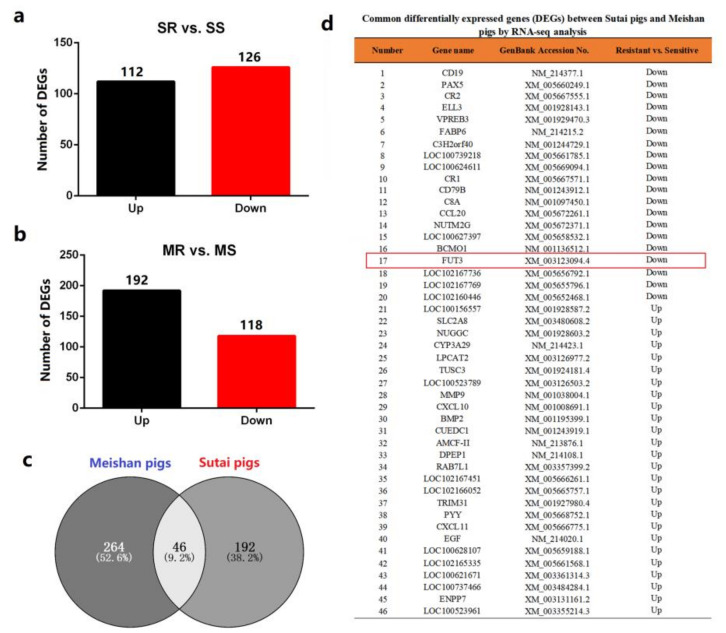
Pig *FUT3* is found as a host gene involved in the infection of *E. coli* F18. (**a**) DGEs between Sutai F18-resistant (SR) and -sensitive (SS) piglets. (**b**) DGEs between Meishan F18-resistant (MR) and -sensitive (MS) piglets. (**c**) Venny screening of common DGEs between Sutai and Meishan piglets. (**d**) Gene list of common DGEs has been associated with infection caused via *E. coli* (F18).

**Figure 2 genes-12-01586-f002:**
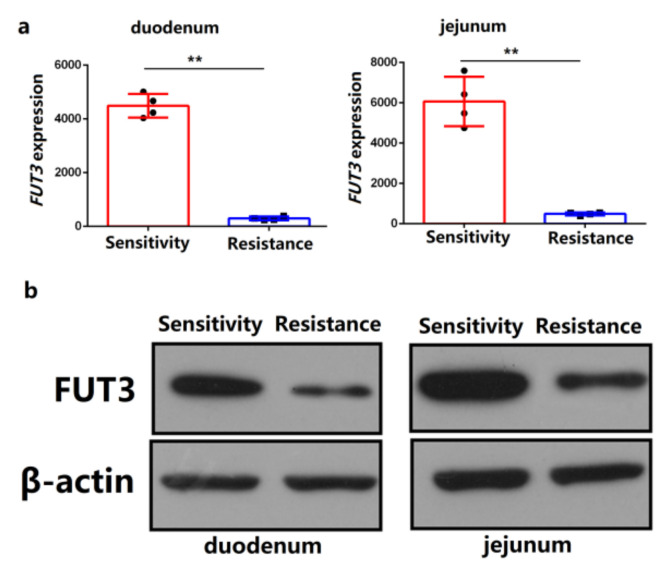
Differential expression analysis of pig *FUT3* gene in intestinal tissues between Sutai *E. coli* F18-resistant and -sensitive piglets. (**a**) qRT-PCR detection, *n* = 3 biological replicates, ** *p* < 0.01. (**b**) Immunoblot analysis, β-actin, internal reference.

**Figure 3 genes-12-01586-f003:**
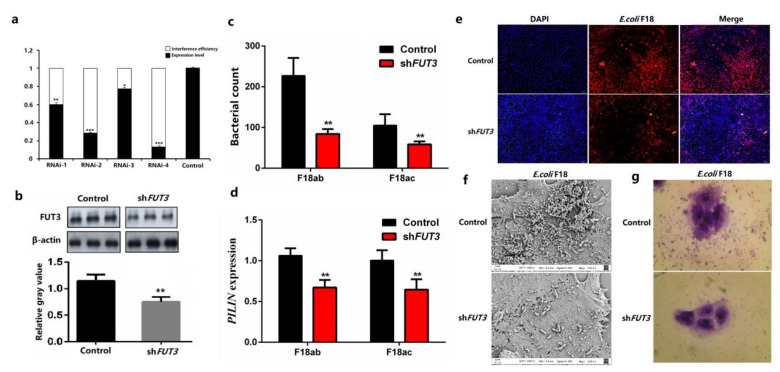
Knockdown of *FUT3* enhances *E. coli* F18 resistance. (**a**) Interference efficiency evaluation of *FUT3* in RNAi-n (*n* = 1, 2, 3, 4) by qRT-PCR. (**b**) Interference efficiency of *FUT3* for RNAi-4 was assessed by immunoblot validation. (**c,****d**) Adhesion of the F18 fimbria to IPEC-J2 cells were evaluated via bacteria enumeration and relative quantification detection, results are represented as mean± SEM, *n* = 3 biological replicates, ** *p* < 0.01. (**e**) Immunofluorescence assay. Blue fluorescence indicates nuclear staining via DAPI, red fluorescence indicates *E. coli* antibody. (**f**) Scanning electron microscopy (SEM) assay, cells were found under a scanning electron microscope (2500×). (**g**) Gram staining assay. An optical microscope (400×) was used to observe cells.

**Figure 4 genes-12-01586-f004:**
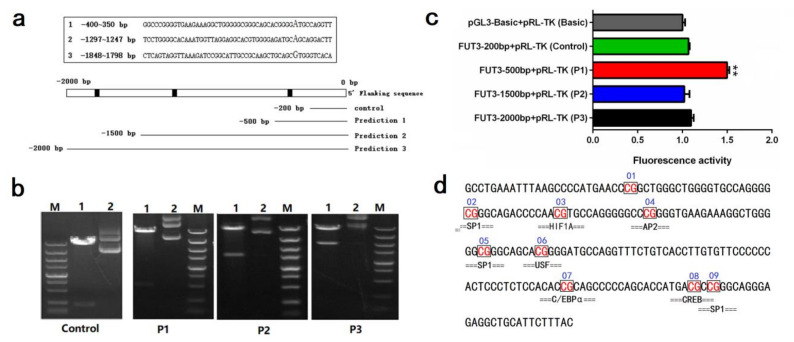
Identification of pig *FUT3* gene core promoter region. (**a**) prediction of *FUT3* core promoter region and truncation of detection fragment. (**b**) Agarose gel electrophoresis of PCR products that were digested by restriction enzymes. Lane 1: Plasmid digested by *Kpn*I and *Hind*III, Lane 2: Plasmid DNA, Lane M: DL5000 Marker. (**c**) Luciferase assay of different vectors. Basic: negative control. Results are indicated as mean± SEM, *n* = 3 biological replicates, ** *p* < 0.01. (**d**) Methylation amplification fragment was designed using the MethPrimer software. Box labeled means the methylated CpG sites, below box means the predicted transcription factors.

**Figure 5 genes-12-01586-f005:**
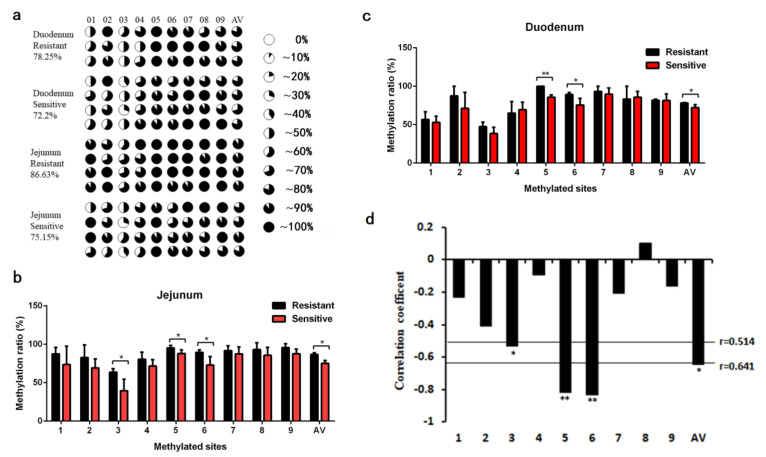
Effect of pig *FUT3* promoter methylation level on gene expression. (**a**) The methylation level of the identified CpG sites. 1~9 mean the different CpG sites, accordingly. CpG sites are observed with pie charts in which the black region shows the level of methylation. AV: the average degree of methylation. (**b**) Differential methylation analysis in jejunum tissues from *E. coli* F18-sensitive and -resistant individuals. (**c**) Differential methylation analysis in duodenum tissues from *E. coli* F18-sensitive and -resistant individuals. Results are indicated as mean± SEM, * *p* < 0.05, ** *p* < 0.01. (**d**) Correlation analysis between the *FUT3* methylation and mRNA expression level.

**Figure 6 genes-12-01586-f006:**
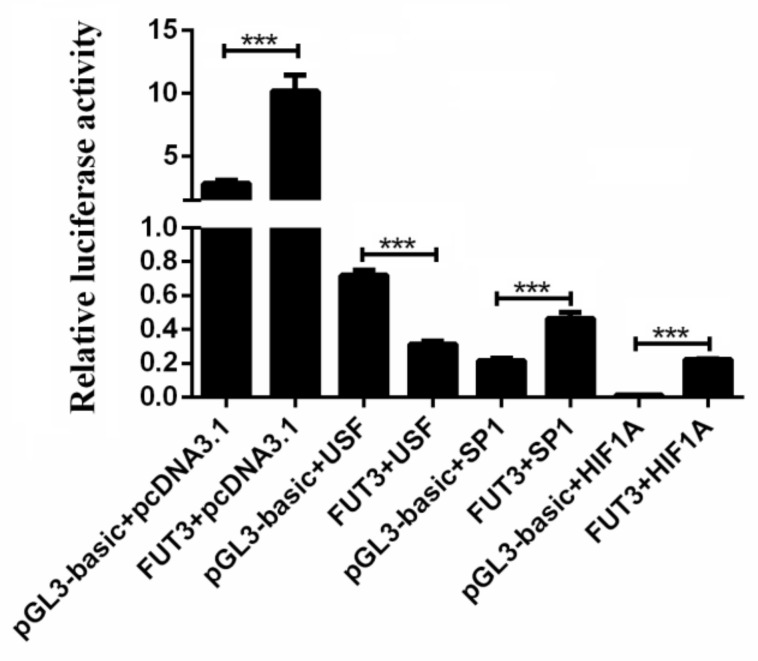
Effects of HIF1A, SP1, and USF on *FUT3* transcriptional activity by dual-luciferase activity assay. The obtained results are indicated as mean± SEM, *** *p* < 0.001.

**Table 1 genes-12-01586-t001:** Primer sequence of pig *FUT3* shRNAs.

Name.	Sequence of the Oligo (5’ → 3’)
R1F	*GATCC*ggtctggttcagcatggaatcTTCAAGAGAgattccatgctgaaccagaccTTTTTTG
R1R	*AATTC*AAAAAAggtctggttcagcatggaatcTCTCTTGAAgattccatgctgaaccagaccG
R2F	*GATCC*gcagggactctgatatcttcaTTCAAGAGAtgaagatatcagagtccctgcTTTTTTG
R2R	*AATTC*AAAAAAgcagggactctgatatcttcaTCTCTTGAAtgaagatatcagagtccctgcG
R3F	*GATCC*gctcaacatctcggccaagaaTTCAAGAGAttcttggccgagatgttgagcTTTTTTG
R3R	*AATTC*AAAAAAgctcaacatctcggccaagaaTCTCTTGAAttcttggccgagatgttgagcG
R4F	*GATCC*gacctccaagtggacgtgtatgTTCAAGAGAcatacacgtccacttggaggtcTTTTTTG
R4R	*AATTC*AAAAAAgacctccaagtggacgtgtatgTCTCTTGAAcatacacgtccacttggaggtcG
NC-F	*GATCC*ttctccgaacgtgtcacgtTTCAAGAGAagttagttgggactttgttgcTTTTTTG
NC-R	*AATTC*AAAAAAttctccgaacgtgtcacgtTCTCTTGAAagttagttgggactttgttgcG

The italic is the introduced enzyme loci, the lowercase is the interference sequence and complementary sequence, and underline represents the loop sequence.

**Table 2 genes-12-01586-t002:** Real-time PCR primers and sequences.

Gene Name	GenBank Accession Number	Primer Sequence	Fragment Size
*FUT3*	AF130972.1	F: 5′-CCCGAAGCCTTCATCCACAT-3′	150 bp
		R: 5′-CATCAAGGCCCAGCTGAAGA-3′	
*PILIN*	M25302.1	F: 5′-AGGCCGAACCAAAGAAGCAT-3′	117 bp
		R: 5′-TCACCATCAGGGTTTCTGAGT-3′	
*β-actin*	NC_010445.3	F:GTCGTACTCCTGCTTGCTGAT R:CCTTCTCCTTCCAGATCATCGC	119 bp

## Data Availability

Transcriptome sequencing data has been submitted to NCBI’s SRA repository under BioProject IDs: PRJNA476718, PRJNA476720, PRJNA476721, PRJNA476722.

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
