# Peer review of "DNA Methylation of Pig FUT3 Promoter Alters mRNA Expression to Regulate E. coli F18 Susceptibility"

_genes, 2021, doi:10.3390/genes12101586_

Round 1

Reviewer 1 Report

In this paper, Authors identify methylated sites in the promoter region of FUT3 gene as possible responsible for the down regulation of expression associated with resistance to piglet diarrhea determined by E.coli F18.

I have no doubts on the results, the several accomplished experiments. It is really a huge laboratory activity under the presentation of the results. However, I do have only some observations to do:

Nothing is said on a simple control. The differences in the sequences of the FUT3 gene in the two analyzed groups. Including flanking regions, of course. Just to be clear, if you don't see any difference, then you add it to the results and nothing changes for the whole context of the paper. Since I am sure that you will see differences, as it might be expected when you analyze some thousands of nucleotides in a sample of, say, 10 individuals, and then results and consequences may change, largely change. Such control could help also in understanding whether you have different levels of resistance.

The second observation is that I find a little difficult to understand how "molecular breeding strategies" can be developed in the case of nucleotide methylation events which could be miles away from FUT3 gene.

Finally, FUT3 gene is only one of the differentially expressed 46. Please, be careful!

Author Response

Dear Editors and Reviewers:

Thank you for your letter and for the reviewers’ comments concerning our manuscript entitled “DNA methylation of FUT3 promoter alters mRNA expression to regulate E. coli F18 susceptibility in weaned piglets” (ID: genes-1377878). Those comments are all valuable and very helpful for revising and improving our paper, as well as the important guiding significance to our researches. We have studied comments carefully and have made correction which we hope meet with approval. The main corrections in the paper and the responds to the reviewer’s comments are as flowing:

Reviewer: 1
In this paper, Authors identify methylated sites in the promoter region of FUT3 gene as possible responsible for the down regulation of expression associated with resistance to piglet diarrhea determined by E.coli F18.

I have no doubts on the results, the several accomplished experiments. It is really a huge laboratory activity under the presentation of the results. However, I do have only some observations to do:

Nothing is said on a simple control. The differences in the sequences of the FUT3 gene in the two analyzed groups. Including flanking regions, of course. Just to be clear, if you don't see any difference, then you add it to the results and nothing changes for the whole context of the paper. Since I am sure that you will see differences, as it might be expected when you analyze some thousands of nucleotides in a sample of, say, 10 individuals, and then results and consequences may change, largely change. Such control could help also in understanding whether you have different levels of resistance.

Response: Thanks for your comments. The two analyzed groups (resistant, sensitive), were obtained by previously challenging with the E. coli through feeding strains i.e., F18ab and F18ac. In fact, we also established a simple control, but we found there was no difference between resistant group and control group. Therefore, we finally selected resistant- and sensitive-piglets for sequencing and methylation, but not selected control group. Although the differences of FUT3 promoter sequences in different individuals, such as single nucleotide polymorphism (SNP), but the core promoter sequence is relatively conservative. In our study, we determined the core promoter region (–500–(–206) bp) of pig FUT3 by bioinformatic prediction and dual luciferase assay. On this basis, we detected the level of methylation in FUT3 core promoter in intestinal tissues of piglets i.e., sensitive and resistant to E. coli F18.

The second observation is that I find a little difficult to understand how "molecular breeding strategies" can be developed in the case of nucleotide methylation events which could be miles away from FUT3 gene.

Response: Thanks for your suggestion. Since we determined the core promoter region (–500–(–206) bp) of pig FUT3, we can perform a screening for mutation sites (SNPs) in the region from different pig breeds. According to our study, we might screen out some SNPs located at methylation sites or transcription factor binding sites, so these SNPs will be used for molecular breeding in future.

Finally, FUT3 gene is only one of the differentially expressed 46. Please, be careful!

Response: Thanks for your suggestion. E. coli F18, which uses its fimbria to attach and subsequently adhere to the brush border F18 receptors of the epithelial cells of small intestine of piglets, resulting in production of enterotoxin, causing diarrhea in piglets (Boldin, 2008). Therefore, resistance against E. coli F18 relies on individual immunity as well as on F18 receptor expression. Recent studies concentrated on the relationship between immunity-related genes and resistance to E. coli F18 in pigs (Wu et al, 2016; Liu et al, 2016; Wang et al, 2014) and a few have shown the formation of the receptor. In this study, we found a differentially expressed gene-FUT3 is involved in glycosphingolipid biosynthesis correlated with the generation of E. coli F18 receptor (Coddens et al, 2009; Moonens et al, 2012; Lonardi et al, 2013), which is probably considered as a novel target to combat E. coli F18 infection in piglets. Of course, other genes of differentially expressed 46 are also important, which will also be paid attention to in future studies.

Boldin B (2008) Persistence and spread of gastro-intestinal infections: the case of enterotoxigenic Escherichia coli in piglets. B Math Biol 70: 2077–2101.

Wu ZC, Liu Y, Dong WH, Zhu GQ, Wu SL, Bao WB (2016) CD14 in the TLRs signaling pathway is associated with the resistance to E. coli F18 in Chinese domestic weaned piglets. Sci Rep 6: 24611.

Liu Y, Gan LN, Qin WY, Sun SY, Zhu GQ, Wu SL, Bao WB (2016) Differential expression of Toll-like receptor 4 signaling pathway genes in Escherichia coli F18-resistant and -sensitive Meishan piglets. Pol J Vet Sci 19: 303–308.

Wang J, Liu Y, Dong WH, Huo YJ, Huang XG, Wu SL, Bao WB (2014) Dynamic changes in TAP1 expression levels in newborn to weaning piglets, and its association with Escherichia coli F18 resistance. Genet Mol Res 13: 3686–3692.

Coddens, A.; Diswall, M.; Angström, J.; et al. Recognition of blood group ABH type 1 determinants by the FedF adhesin of F18-fimbriated Escherichia coli. Journal of Biological Chemistry. 2009, 284, 9713–9726.

Moonens, K.; Bouckaert, J.; Coddens, A.; et al. Structural insight in histo-blood group binding by the F18 fimbrial adhesin FedF. Molecular Microbiology. 2012, 86, 82–95.

Lonardi, E.; Moonens, K.; Buts, L.; et al. Structural sampling of glycan interaction profiles reveals mucosal receptors for fimbrial adhesins of enterotoxigenic Escherichia coli. Biology (Basel). 2013, 2, 894–917.

Reviewer 2 Report

Dear Authors,

You have presented some very interesting research in the manuscript entitled: DNA methylation of pig FUT3 promoter alters mRNA expression to regulate E. coli F18 susceptibility.

My only substantive comment is the short-term of the FUT3 gene silencing effect presented in the results. I would like to point out that a very worthwhile experiment would be the one you mention in the discussion - the CRISPR/Cas knockout experiment. Perhaps it is worth obtaining a modified animal to check the described results in vivo?

Nevertheless, I positively evaluate the presented work. I have one editorial note - in Figure 3B the descriptions of the bars are very, very small and practically invisible - please enlarge this to make it clearer.

Best regards.

Author Response

Dear Editors and Reviewers:

Thank you for your letter and for the reviewers’ comments concerning our manuscript entitled “DNA methylation of FUT3 promoter alters mRNA expression to regulate E. coli F18 susceptibility in weaned piglets” (ID: genes-1377878). Those comments are all valuable and very helpful for revising and improving our paper, as well as the important guiding significance to our researches. We have studied comments carefully and have made correction which we hope meet with approval. The main corrections in the paper and the responds to the reviewer’s comments are as flowing:

Reviewer: 2

You have presented some very interesting research in the manuscript entitled: DNA methylation of pig FUT3 promoter alters mRNA expression to regulate E. coli F18 susceptibility.

My only substantive comment is the short-term of the FUT3 gene silencing effect presented in the results. I would like to point out that a very worthwhile experiment would be the one you mention in the discussion - the CRISPR/Cas knockout experiment. Perhaps it is worth obtaining a modified animal to check the described results in vivo?

Response: Thanks for your valuable suggestion, your comment is very constructive. To evaluate whether FUT3 affected E. coli F18 resistance in vivo, we are examining the susceptibility to E. coli F18 of mice as model animals. In normal mice, we have determined the optimal dose (1.0×1010 CFU/mL) of E. coli F18 infection using survival rate analysis. Then have got the Fut3 knockout cell line. And the next step we plan to construct the Fut3 condition knockout in intestinal mouse model. As the condition knockout mouse model need to make the Fut3fl/- mouse, which maybe need six months, and the Fut3-/- mouse were from the CRE mouses and Fut3fl/ mouses, which maybe need 8 months. Furthermore, the Fut3-knockout (Fut3-/-) mouse model will be established by PCR detection, sequencing, and western blotting analyses. Wild-type (Fut3+/+) and Fut3-/- mice will be challenged with 1.0 × 1010 CFU/mL dose of E. coli F18. 

Nevertheless, I positively evaluate the presented work. I have one editorial note - in Figure 3B the descriptions of the bars are very, very small and practically invisible - please enlarge this to make it clearer.

Response: Thanks for your correction. As requested, we enlarge Figure 3B (it was modified to Figure 3a in revised manuscript) to make it clearer.

Round 2

Reviewer 1 Report

After the answers of the Authors, the only thing I can say is that in case of phenotypic differences, you first search for genetically determined differences and if you don't find them, then you can talk about epigenetics. Making the reverse is a nonsense (as stated in the answer n.2) since you will always have the doubt that the observed genetic differences are causative of the different phenotypes rejecting the epigenetic origin.

I think that any reference to epigenetics in the present form of the paper should be eliminated. 

Author Response

After the answers of the Authors, the only thing I can say is that in case of phenotypic differences, you first search for genetically determined differences and if you don't find them, then you can talk about epigenetics. Making the reverse is a nonsense (as stated in the answer n.2) since you will always have the doubt that the observed genetic differences are causative of the different phenotypes rejecting the epigenetic origin.

I think that any reference to epigenetics in the present form of the paper should be eliminated.

Response: Thanks for your comment. For bacterial diarrhea, we obtained different phenotypic piglets with resistance and sensitivity to E. coli F18. Using transcriptome sequencing, we analyzed genetic differences between F18-resistant and -sensitive individuals, and identified some candidate genes, including FUT3. In our study, we firstly confirmed that the upregulated expression of FUT3 enhances E. coli F18 susceptibility. On this basis, we further detected the DNA methylation of FUT3 promoter, which was used to explore the effect of DNA methylation on FUT3 expression, but not phenotypic differences. Our study aimed to reveal the regulatory mechanism of DNA methylation on FUT3 expression. I apologize for not making it clear in the paper, we made some appropriate modifications in the introduction.
